# Illiberal Cultural Christianity? European Identity Constructions and Anti-Muslim Politics

Anja Hennig [1] and Oliver Fernando Hidalgo [2,*]

1   Faculty of Social and Cultural Sciences, European University Viadrina, 15230 Frankfurt (Oder), Germany; ahennig@europa-uni.de
2   Institute for Political Science, University of Münster, 48149 Münster, Germany
*   Correspondence: hidalgoo@uni-muenster.de

**Abstract:** This paper refers to the ambivalence of secularization in order to explain why Cultural Christianity can show both a liberal and illiberal character. These two faces of Cultural Christianity are mostly due to the identity functions that, not only faith-based religion, but a particularly culturalized version of religion, entails. Proceeding from this, it will be demonstrated here how Cultural Christianity can turn into a concrete illiberal marker of identity or a resource for illiberal collective identity. Our argument focuses on the link between right-wing nationalism and Cultural Christianity from a historical-theoretical perspective, and illustrates the latter with the example of contemporary illiberal and selective European memory constructions including a special emphasis on the exclusivist elements.

**Keywords:** Cultural Christianity; illiberalism; ambivalence; secularization; democracy; European identity; memory politics

## 1. Introduction

In 2014, the appearance of the German right-wing movement, "Patriotic Europeans Against the Islamization of the Occident" (known as PEGIDA), in Eastern Germany came as something of a surprise—not least because the term "Occident" evokes "Judeo-Christian culture", and Eastern Germany is not a particularly religious part of the country. Frequently, during its weekly marches through the city center of Dresden, someone would be carrying a wooden cross painted black, red, and gold, the colors of the German flag. Most of PEGIDA's adherents, however, not only do not belong to any Christian denomination, but they are actually critical of the mainstream churches, while the churches themselves keep a strict distance from this "perversion" of Christian dogma (Resing and Orth 2017).

The German PEGIDA movement has since become a marginal, albeit not irrelevant, phenomenon. What it illustrates, however, is a Europe-wide trend within nationalist anti-Muslim rhetoric towards mobilization against the immigration and accommodation of people with Muslim backgrounds by construing national and European belonging as fundamentally Christian. Toward this end, and following Marzouki et al. (2016), nationalists "hijack" religion for their illiberal purposes. When churches speak about faith, right-wing populists speak about identity.

Given the historically constructed antagonism between "Orient" and "Occident", it seems plausible to consider the right-wing populist invocation of Christianity as a pragmatic strategy that construes Islam as "the other", as the "enemy" of Europe (Brubaker 2017; Joppke 2018). Nonetheless, it remains astonishing that, in times of decreasing religiosity and increasing distance from Christian traditions, religion is now experiencing such an unholy renaissance throughout Europe—all the more so given that a culturalized version of Christianity is nowadays associated with the liberal accommodation of (civilized) religion[1] rather than with the illiberalism we see today. Hence, we are taking this puzzling development as a point of departure to explore, theoretically and historical-empirically

from a political science perspective, the illiberal degeneration and appropriation of what we conceptualize as "Cultural Christianity".

Literature that observes a trend towards the culturalization of Christianity is not new. In the 1990s and early 2000s, questions as to how religion, and more specifically religiosity, were transforming in the context of secularization and cultural pluralization were discussed, primarily among sociologists of religion. For instance, Jagodzinski and Dobbelaere (1993) noticed a particular trend towards a kind of sociocultural Christianity, here understood as an effect of the erosion of confessional pillars. Instead of the Catholic pillar, they saw the evolution of a "Christian pillar" (Jagodzinski and Dobbelaere 1993, p. 76). In 2000, Demerath very clearly addressed the phenomenon of "cultural religion", which he saw primarily as an effect of secularization. The same year, the French sociologist of religion, Danièle Hervieu-Léger, published her seminal book, *Religion as Chain of Memory* (Hervieu-Léger 2000), in which she elaborated the idea of a "religious belonging without believing".

The overarching question of the above-mentioned approaches concerns the roles that religion, and Christian religiosity in particular, still play in a secularizing environment. Steve Bruce (2002, pp. 39–46) identified one of these roles as the phenomenon of "cultural defense". According to Bruce, religion in general (and in the secularized Western world, Christianity in particular) can be successfully functionalized to defend one's own culture, especially at moments when the need to shore up a religious-secular identity increases as a defensive reaction against a widespread sense of alienation, rootlessness, and foreign infiltration in large parts of the population.

With the rise of right-wing movements and parties, especially after the financial crisis of 2008, social scientists took a more critical, problem-solving approach to the heretofore rather neutral analysis of the trend that saw religiosity moving towards a nonreligious sense of Christian belonging. For instance, Marzouki, McDonnell, and Roy (2016) reveal in their edited volume how "populists hijack religion" to mobilize against Islam. The edited volume on illiberal politics and religion from Hennig and Weiberg-Salzmann (2021) confirms this unholy mechanism and underlines how the illiberal political appropriation of Christianity occurred rather slowly over time. In a similar vein, but with a regional focus on Western Europe, Brubaker (2017, p. 1193) identified a "shift from [old-fashioned] nationalism" to a new kind of "civilizationism" as a paradoxical combination of identitarian Christianity, secularism, philo-Semitism, Islamophobia, and, finally, "an ostensibly liberal defence of gender equality, gay rights and freedom of speech", which are celebrated (and misinterpreted) as exclusive achievements of the Christian World against the alleged backwardness of Muslim culture. Joppke (2018), too, refers to right-wing actors, seeing them, however, merely as driving forces towards the culturalization of religion. A pathbreaking contribution that substantially systematizes the panorama of social science research on what the authors call "culturalized religion" comes from Astor and Mayrl (2020). They provide a conceptualization of three modalities of culturalized religion that is consequential for our paper as well: constituted culture, pragmatic culture, and identity.

Despite the new intensity within research on culturalized religion, a more systematic conceptualization of Cultural(ized) Christianity, and the question as to how such a sociological phenomenon or sociocultural construct can be used for illiberal purposes, is still lacking. Against this background, our paper aims at approaching what we conceive of as Cultural Christianity from a conceptual and historical-theoretical perspective. It seeks to provide analytical pathways in order to explain, to a certain extent, why Cultural Christianity—as a discursive frame, a sense of belonging, or a reservoir for constructing cultural heritage or tradition—is vulnerable to appropriation in an illiberal way.

Our main argument is threefold. First, it is the ambivalent character of secularization that generally explains why any reference to Cultural Christianity can occur in a liberal sense, while also being simultaneously vulnerable to use for illiberal purposes. Second, this ambivalent character of Cultural Christianity has largely to do with the identity functions that, not only faith-based religion, but a particularly culturalized version of religion,

entails. Therefore, the aim of Section 2 is to conceptualize our understanding of Cultural Christianity, with its illiberal potential, and to clarify the theoretical concepts relevant to our argument, such as secularization and illiberal politics. Section 3 then suggests two main social mechanisms through which Cultural Christianity can turn into a concrete illiberal marker of identity or resource for illiberal collective identity construction, namely, as demand- and supply-side factors for right-wing actors. The third and last aspect of our argument focuses on the link between nationalism and Cultural Christianity. From a historical-theoretical perspective focusing on nationalism, Cultural Christianity—as a merely superficial form of religiosity, indifferent to faith and theological knowledge—is particularly susceptible to such selective instrumentalization by right-wing actors, who deliberately obscure what is actually an obvious contradiction between the territorial logic of the nation and the universalist logic of Christianity. Proceeding from this, Section 4 deals with the example of contemporary illiberal and selective European memory constructions, with their increasing emphasis on exclusivist elements. We will show how right-wing nationalists today fuse the secular idea of the modern nation with the earlier idea of Christian Europe in order to reanimate the distinction between Occident and Orient and, thus, to push the intellectual exclusion of Muslim immigrants.

## 2. Cultural Christianity and the Ambivalence of Secularization

For more than two decades, the sociology of religion has been observing a trend towards a differentiation between faith and the cultural dimension of religion. The aim of this section is, therefore, to put the rise of culturalized religion into context in order to clarify the meaning of Cultural Christianity. To begin with, culture has always influenced, and always been part of, religion. According to Roy (2010), religion creates culture because religion is also lived as culture. There are countless pertinent examples of religious cultural artifacts. In order to understand which factors affect the relations between religion and culture, the first part of this section introduces the debate surrounding culturalized Christianity, which, as the second part reveals, is linked to what we call ambivalence of secularization processes.

### 2.1. Conceptualizing Cultural Christianity

Within the conceptual debate concerning the phenomenon of a cultural, rather than a religious, understanding of Christianity and its evolution and relevance in the present, one can distinguish two major interrelated directions. The first concerns whether the term is conceived of as neutral or as ideologically loaded, while the second concerns the specificity of the actors the term might be presumed to encompass, their scope, and the differences between countries. The latter also concerns the question as to whether actual religious actors can themselves rely on cultural Christian(ist) narratives (Ryan 2021). In this context, we depart from a neutral and broader systematic understanding (Meulemann 2019) to more narrowly define Cultural Christianity's vulnerability to illiberal appropriation as an ambivalent effect of secularization. To that end, we rely on the threefold conceptualization Astor and Mayrl (2020) elaborate in their sociological review essay. The authors propose the term "culturalized religion" to understand the contemporary trend towards a cultural, rather than a faith-based, affiliation with religion. Astor and Mayrl analytically differentiate between three distinct, yet interdependently functioning, modalities of culturalized religion. These are a *constituted* culture, a *pragmatic* culture, and an *identity-related* culture. Constituted culture captures how religion may become deeply embedded as "deposits of faith" in institutions and individuals in ways that are not necessarily recognized as religious. Pragmatic culture, by contrast, captures instrumental forms of culturalized religion. Here, actors define aspects of religion as "culture" in—and this is important for our perspective—the service of concrete political projects. One example would be the defense of employing religious symbols as cultural symbols in the public sphere, like PEGIDA did with the flag-colored cross. The identity mode captures those forms of culturalized religion that are used as, or have become, markers of communal belonging understood

in religious terms, albeit in the absence of broader religious participation or salience (Astor and Mayrl 2020, p. 213). Here, the invocation of Europe's Christian heritage could serve as an example.

Whereas this systematization opens the door for analyses of various uses (and abuses) of culturalized religion, the term "Christianism"—coined by Brubaker and more thoroughly conceptualized by Ryan—is used in an ideologically narrow sense to refer to illiberal actors that appropriate Christianity against Islam specifically (Brubaker 2017; Ryan 2021). Christianism, from this point of view, is a clearly secular phenomenon of the European West. For Brubaker (2017, p. 1199), it is "a civilizational and identitarian Christianity", being neither "substantive" nor substantial and, therefore, able to cope with secular culture. In this context, Brubaker adopts the idea of a "secularized Christianity-as-culture" from Mouritsen (2006, p. 77) and associates it with Roy (2016) and the matter of defining "us" in relation to "them". These intellectual sources lead Brubaker (2017, p. 1199) to secular Christianism/culturalized Christianity's "crude" conclusion: "If 'they' are Muslim, 'we' must, in some sense, be Christians"—which does not involve or demand one be(come) "religious" in the strict sense of the word but only "invoke[s] Christianity as a cultural and civilizational identity".[2]

Ryan explicitly refers to Brubaker in this respect. However, Ryan's critique of the "empty [identitarian] symbolism" of Christianism apparently treats the latter as a pure "political ideology" (Ryan 2018) by suggesting it is a "deliberate mirroring of the term *Islamism*" (Ryan 2021) that has nothing to do with religion. In contrast to this, we rather insist on the intersections of Christianism, Cultural Christianity, and the cultural heritage of the Christian past. We will come back to this when discussing the susceptibility to illiberal use of what we call Cultural Christianity.

In our understanding, Cultural Christianity is a version of culturalized religion that is clearly shaped by a traditional majority religion. As do Davie (1990) and Meulemann (2019), we are referring to both a neutral sense of self-identification with Christian traditions, institutions, values, or rites, as well as an instrumental use of references to a Christian collective identity, such as the construction or selective remembrance of a Christian heritage. This sort of unspecified sense of belonging to a Christian tradition or institution, without being observant, is one marker of the contemporary ambivalence of modernity and Europeanization.[3] Whereas most references to what is perceived as the Christian tradition, or the "faith-less" use of Christian symbols or celebrations of Christian holidays, are the ordinary ingredients of secularizing societies in liberal democracies (Meulemann 2019, p. 268f.),[4] when seen through the lens of the ambivalence of secularization, as we propose in the following sections, Cultural Christianity also increasingly bears the potential to function in an illiberal sense.

*2.2. The Ambivalence of Secularization*

We share the assumption that the emergence of culturalized versions of Christianity is an effect of secularization. Following Demerath (2000, p. 136), "cultural religion may represent the penultimate state of religious secularization—the last loose bond of religious attachment before the ties are let go altogether. In many societies around the world—and perhaps especially in Europe—cultural religion may represent the single largest category of religious orientation. It is a self-applied label. It is a way of being religiously connected without being religiously active. [ ... ] It is a tribute to the religious past that offers little confidence for the religious future".

To understand why culturalized religion can function in both a liberal and an illiberal way, we take the ambivalent effects of secularization into consideration. This metaperspective emphasizes the ambivalences of contemporary developments that are rooted in processes of modernization, secularization, and democratization. Bauman (1991) and Beichelt et al. (2019, p. 9) remind us that "modernity and modernization are based on potentially destructive elements. The actual scope of destruction and construction may depend on the question of whether and how violence and power may be distributed, transformed and institutionally channeled. The becoming of Europe comprises remarkable

examples in both directions, thereby indicating that the ambivalences of modernization translate into the ambivalences of "Europeanization", where "Europeanization" is understood as the assumed or normatively desired processes of convergence in political, cultural, or economic terms.

In an epistemological sense, ambivalence implies here a nonteleological and less normative view of the dynamics between liberal-democratic consensus building and countermovements. In the modern era, Europe has generally served as a major battleground for relevant contradictions. Colonial history, the battles between church and state, the legacy of Catholic or "secular" nationalisms and anti-Judaism, and the rise of the radical right in the 1980s in Western Europe and the United States, do not simply belong exclusively to the past. Through the lens of ambivalence, these stories can be seen as forming part and parcel of the liberal project (Hennig and Hidalgo 2021, p. 37ff). In a similar vein, secularization processes also show their ambivalent character; there is, on the one hand, an increasing public visibility of religion through, for instance, demands made by religious minorities or the reference to religious traditions, and, on the other, a decrease or transformation in religiosity and religious practice (see e.g., Casanova 1994; Berger 1999; Habermas 2003; Roy 2010; Zapf et al. 2018). Here, Habermas' notion of the "post-secular" condition provides ground for both a reclaiming of religious identities in the public sphere, and the decreasing relevance of religious beliefs and practice (Hennig and Hidalgo 2021, p. 50).

These ambivalence of secularization processes are already due to the concept of the secular itself. Although, broadly speaking, the concept of the secular, or secularization, "anywhere in the world, means a separation of organized religion from organized political power, inspired by a specific set of values", it is not a "doctrine with a fixed content" but "has multiple interpretations which change over time" (Bhargava 2010, p. 65; see also Martin 1978, 2005, or Eisenstadt 2003). Moreover, this fundamental differentiation of politics and religion in a secular state or society—which at least theoretically ends all religious hegemony, intrareligious oppression, and inter-religious domination—should not be confused with a completely apolitical role for religion and a denial of its genuinely political dimensions (Casanova 1994, 2008). Rather, secularism or secularization does not require a strict separation between the political and religious spheres, as such, but can be understood as a concept that marks a specific change, as well as an arrangement, within the relationship between the two spheres. In this respect, the main characteristic of the secular can be recognized in the permanent *coexistence* of religious and nonreligious values, lifestyles, and options (Taylor 2007). Conceptualizing Cultural Christianity as an outcome of secularization, as we propose in this article, therefore means that it can be seen as a phenomenon that is simultaneously religious *and* nonreligious.

Olivier Roy demonstrates very well the ambivalent effects of secularization on the transformation of the relations between religion and culture. On the one hand, secularization affects faith but not necessarily values. Therefore, a secularized society can remain in step with religious culture and values. Secular and religious people can both agree on religious values. At the same time, when those who do profess faith in a religion no longer identify with the contemporary surrounding culture, and when this surrounding culture no longer accepts religion, something happens that Roy (2010, p. 113), following Hervieu-Léger, calls "exculturalization". This exculturalization of religion, which here demonstrates another facet of the mentioned ambivalence, is a key development in present-day globalization, and it largely explains the success of fundamentalist forms of religion fighting against their secular environments, which they perceive as decadent. In other words, the exculturalization of religion occurs when the religious norm breaks away from (secular) culture. For religion, culture suddenly appears pagan, and no longer merely a profane or secular reality (Roy 2010, p. 115).

In the sociology of religion, as early as the 1990s and early 2000s, two hypotheses, which initially appeared to be in competition with one another, had already grasped the ambivalent dynamic between a stronger sense of Christian identity and a decrease in religiosity: Grace Davie's "believing without belonging", which considers Christianity a

marker of identity (Davie 1990; PEW 2005), and Danièle Hervieu-Léger's reverse idea of a "belonging without believing" (Hervieu-Léger 2000). To quote Casanova (2006, p. 67), "large numbers of Europeans even in the most secular countries still identify themselves as 'Christian', pointing to an implicit, diffused, and submerged Christian cultural identity. In this sense, Hervieu-Léger is also correct when she offers the reverse characterization of the European situation as 'belonging without believing'. 'Secular' and 'Christian' cultural identities are intertwined in complex and rarely verbalized modes among most Europeans."

Within the context of the ambivalent effects of secularization following modernization, two more specific factors are considered relevant for the rise of Cultural Christianity in liberal democracies. According to Joppke (2018, pp. 4–6), one of the two important forces driving the culturalization of religion is "radical right parties that depict a *Christian* Europe as threatened by immigrant Islam". (Section 3.2 will analyze this point from a supply-side perspective.) The second factor is the legal system, and the particular role that courts play in the culturalization of religion: they face the problem of "justifying an inevitably privileged position of majority religion in a historically Christian society". At the same time, courts also have to respect the liberal principle of state neutrality (Joppke 2018, p. 3). To consider an originally religious symbol, such as the crucifix, in the public sphere not as a religious, but a cultural expression of identity, would be such a strategy of "culturalization", or legitimizing religious expression not being objected to by public courts as a violation of state neutrality (Astor and Mayrl 2020, p. 215). In more general terms, Burchardt (2020, p. 155) concludes that "references to religion as heritage are an often-overlooked consequence of secularization that does not so much lessen as heighten the role of religion in a nation-state".

## 2.3. The Ambivalence of Democracy and "the Illiberal"

Having conceptualized our understanding of Cultural Christianity and introduced ambivalence as a metaperspective on secularization, the final step within this first theoretical section is to conceptualize "the illiberal". Here, we rely on an understanding of "illiberal politics" that characterizes actors and ideas striving to enact anticonstitutional, antipluralist, and exclusivist policies (Minkenberg 2018). The analytical purpose of "the illiberal" is to characterize or identify (political) agendas and processes that aim at contradicting or opposing liberal-democratic principles, such as the rule of law, the separation of powers, civil rights, or an independent judiciary (Hennig and Hidalgo 2021, p. 41).

In most circumstances, however, and although they generally tend to abolish the normative contradictions that are constitutive for liberal democracy, illiberal reasons and purposes might remain permanently *within* the framework of democracy. In contrast to extremist or radical right- or left-wing political positions, the illiberal shows a Janus face towards democracy that impedes profound analysis of the rather phantom menace that illiberal politics involves. This is how illiberal actors are able to find several arguments to distinguish themselves from extremist and antiliberal enemies of democracy.

Nevertheless, we insist on a concept of the "illiberal" as an antagonistic attribute of democracy that stands in dynamic opposition to its "liberal" elements. Linking back to the idea of ambivalence as a meta-condition, elaborated in Section 2.2, such "liberal/illiberal" opposition can also be found in contemporary cultural spheres such as religion. It is a commonplace, in fact, that religious values, convictions, and identities can function as both a support for liberal institutions, morals, and customs, and as a resource to strengthen illiberal programs, agendas, and arguments. With regard to the first aspect, there are not only countless references concerning the prospects and conditions of successful religious accommodation in the liberal state (e.g., Rosenblum 2000; Stepan 2001; Mookherjee 2011; Beaman 2012; D'Costa et al. 2013; Seglow and Shorten 2019), but also theoretical and empirical evidence that religion can be a positive factor in building social capital and political engagement in liberal democracies (e.g., Bellah et al. 1985; Casanova 1994; Walzer 1998; Putnam 2000; Böckenförde 2013). On the other hand, a growing amount of literature emphasizes the nexus between religion and illiberal politics

(Almond et al. 2003; Marx 2003; Micklethwait and Wooldridge 2009; Hamid 2014; Hibbard 2015; Grzymala-Busse 2015; Marzouki et al. 2016). Thus, religion also entails an ambivalence that is contingent upon the particular actors and political conflicts at stake (Hennig 2018).

To analyze Cultural Christianity's affinity with illiberal attitudes means to identify patterns where "Christianity" is appropriated in such a way as to go against the essential principles of the liberal project, such as freedom of thought and expression, religion, movement and association, and sexual orientation and lifestyle. The liberal project adheres to the idea that the exercise of any particular freedom is to be respected and upheld only insofar as it does not violate the equal freedom of others (Charvet and Kaczynska-Nay 2008). The vulnerability of Cultural Christianity to illiberal ends is thus clearly seen in linkages between individual, exclusivist, and socioculturally homogenizing freedoms that limit the ideas and actions of others.

## 3. Exploring the Illiberal Vulnerability of Cultural Christianity

In Section 3, we aim at identifying the analytical pathways of an illiberal appropriation of Cultural Christianity. Our main argument is that the function of religion—to provide an identitarian sense of individual as well as collective belonging—is central also as an illiberal reference to a culturalized version of, in our case, Christianity. The above-mentioned pragmatic-instrumental and identitarian-communal modes of culturalized religion, as elaborated by Astor and Mayrl (2020), can capture both dimensions: individual identity and identity politics. We will approach the individual dimension from a demand-side perspective, and the identity politics dimension from the supply side. In our view, these different perspectives reveal mechanisms that can explain the vulnerability of Cultural Christianity to a certain extent.

Explanations focusing on the demand side refer to macrostructural factors that have changed the interests, emotions, identities, attitudes, and preferences of voters and adherents of, for instance, political parties. Explanations emphasizing supply-side factors concentrate on what actors are offering in terms of political programs or ideology, party organization, and structures of political opportunity, such as support by partisan media or governmental elites. These analytically separate supply- and demand-side factors can, however, be readily combined (Rydgren 2007, pp. 247, 252). Social changes on the macro level, such as the processes of globalization and migration, have the potential to be perceived by individuals as threats, and may facilitate, for instance, the construction of Islam as a scapegoat by supply-side nationalist actors. National contexts, in addition, determine how, whether, and "which types of right-wing organizations with different ideological tendencies mobilize around different issues" (Caiani 2017, p. 7).

### 3.1. Individual Belongings

Following Astor and Mayrl, culturalized religion as identity should be understood as a type of personal self-understanding linked to collective identity that explains particular forms of social, political, or economic activity (Astor and Mayrl 2020, p. 217). This section discusses empirical data that tries to measure self-identification with Christian culture without a person being explicitly religious; this is an identity demand, which can also be understood in some way as an expression of a reaction to the subjective feeling of increasing uncertainties resulting from globalization, the erosion of traditional patriarchal structures, or the above-mentioned increasing cultural pluralization.

Empirical studies which quantify one's self-understanding as a Christian are still rare (see e.g., Storm 2011).[5] Yet, two comprehensive Pew Research Center surveys, one on Western Europe and the other on Central and Eastern Europe, confirm a sense of belonging, especially to a Christian culture (however defined), that is not correlated with religious behavior or practice. In Western Europe, most adults still *do* consider themselves Christians, even if they seldom go to church. Indeed, the survey shows that nonpracticing Christians (people who identify as Christians but attend church services no more than a few times

per year) make up the biggest share of the population across the region. In every country except Italy, the unaffiliated Christians outnumber the church-attending ones (those who go to religious services at least once a month). The Pew research study concludes that Christian identity remains, despite a large number of unaffiliated Christians, a meaningful marker in Western Europe.[6] The religious, cultural, and political views of this group often differ from those of church-attending Christians and religiously affiliated adults (PEW 2018a, May 29). Also, large parts of the population in Central and Eastern Europe embrace religion as an element of national belonging even though they are not highly observant.

Most interesting for our purposes is that, in both regions, an association between religious and national, if not nationalist, identity exists. A majority of the Eastern Europeans surveyed agreed with the statement, "Our people are not perfect but our culture is superior".[7] The link between Christian identity and nationalism, however, is stronger in Orthodox-majority countries than in Catholic ones (PEW 2017, October 5). In Western Europe, too, on average, a substantial number (54% of church-attending Christians, 48% of nonpracticing Christians, and 25% of religiously unaffiliated people) agree with the idea of cultural superiority. Moreover, the Pew research study finds that, in Western Europe, Catholics are more likely than Protestants to express negative views of Muslims (PEW 2018a, May 29, p. 19; see also Wood 2016, p. 13).

It is important to note, however, that the data on Christian self-identification also includes those who do go to church. The question, though, is to what extent a slightly religious or nonreligious social sense of belonging to Christianity, or to a diffuse Christian culture, is related to the intermingling of Christian religion and illiberal politics. A recent and more nuanced study of the relevance of religion for social identity construction in Germany and Switzerland found that, in both countries, about 11% of nonreligious people with a religious affiliation consider the Christian religion to be important or very important for their social identity. The authors relate this finding to a rather new type of in-group/out-group dynamic, whereby the perception of Islam as "the other" triggers in nonreligious people the conception of their own counteridentity as "cultural Christians" (Liedhegener et al. 2019). Brubaker uses the term "reactive Christianity" to describe this mechanism.[8] In a similar vein, the Bertelsmann Foundation finds that the "rapid transformation of Islam in Europe" is "triggering a reflection on Christian identity also among groups with no religious affiliations" (El-Menouar 2017).

The Pew surveys also found that it is not only in Eastern Europe that large parts of the population embrace religion as an element of national belonging even though they are not highly observant. There also exists in Western Europe an association between religious and national, or partly nationalist, identity (PEW 2017, October 5, p. 8; PEW 2018b, October 29). The data reveal that in Western Europe, on average, 48% of nonpracticing Christians agree with the idea of national cultural superiority, and 37% feel that immigration levels should be reduced (PEW 2018a, May 29, pp. 14/19).[9]

The overall development in the United States is different, where the number of "nones"—meaning unaffiliated and religiously uninterested Americans—has been increasing over the last two decades (PEW 2019, October 17). At the same time, however, Christian nationalism, or the reference to Christian tropes for nationalist politics, is an established and well-known phenomenon closely connected to the Christian right and to the presidency of Donald Trump (see Bechert in this volume). The authors of the Pew report on Western Europe bear in mind that the survey neither fully proves an increase in Christian identity in Europe, nor that, "if Christian identity is growing, immigration of non-Christians is the reason". They do, however, assume that in Western Europe the Christian religion is strongly associated with nationalist sentiments (PEW 2018a, May 29, p. 15).

In sum, it seems safe to say that Christianity can be a reference for social self-identification, which, because of its potential relevance for someone's identity, is vulnerable to mobilization for illiberal politics (and not only by targeting Muslims), irrespective of denomination, religiosity, or the strength of a right-wing party. Whether people construct a

new exclusivist Christian identity as a reaction to Muslim immigration, or because they come from a nationalist Catholic tradition (Storm 2011) is, here, of secondary importance.

### 3.2. Mobilizing Cultural Christianity around Identity Policy Issues

If a certain degree of cultural belonging to Christianity can be considered a condition or demand-side factor for illiberal actors (Rydgren 2007, pp. 247, 252), the supply-side perspective looks at actors, and their programs or strategies, that may make strategic use of the above-mentioned predispositions to develop an exclusivist Cultural Christian identity. In conceptual terms, the supply-side perspective comes close to what Astor and Mayrl identify as pragmatic culturalized religion. This is where "framings of religion as culture abound in discourses on the nation and its heritage", and where the construction of culturally religious identities can have a nationalistic undertone (Astor and Mayrl 2020, p. 216). Political actors may use "religion-to-culture strategies" to allow for political appeals to both observant and nonobservant sections of the population. On the other side, they can rely on culturally religious identities as politically useful resources "for the generation and mobilization of civilizational discourses" or "reactionary identities". The supply side is, thus, about "mobilizing culturalized religion for political purposes" (Astor and Mayrl 2020, p. 221).

To return to the example of PEGIDA, radical right movements, such as the Identitarian movement, or right-wing parties such as the German AfD, the Dutch Vlaams Blok, Fidesz in Hungary, or the Lega Nord in Italy, are precisely actors who push for a Christian identity for European state societies. This development on the political right stands in sharp contrast to the refusal of the European Union to include a reference to Christianity in its never-realized draft constitution (Joppke 2018, p. 4).

This example reveals more specifically how the mobilization of a sense of Cultural Christianity is related to a form of identity politics. It is not distributive policies, such as tax regulation or social welfare, that are the primary arenas of religious-political cooperation (Lowi 1964, 2011). At stake are regulative policies connected to the central principles of the liberal project. Taking the importance of immigration control for nationalist politics into consideration, the regulation of immigration is also a central area of collective identity policy. In a similar vein, one can consider legal litigation about religious symbols in the public sphere, and the question of what constitutes a national heritage, as key questions of identity politics. In this context, "cultural religion" is, according to Demerath (2000, p. 136), "a matter of continuity with generations past and contrast with rival groups and identities. Identity is pivotal to the phenomenon, for religion has always ranked with ethnicity, nationality and social class as salient marker of personhood."

In more general terms, Fukuyama (2018) considers the rise of (collective) identity politics a result of the ideologically driven shift in global politics from the economic left-wing axis to an identity axis, to "fixed characteristics that link us to certain groups", usually based on ethnicity, race, religion, or gender. In our context, right-wing populist identity politics are based on an exclusionary idea of an antipluralist, often nationalist, collective identity (Müller 2019). From this perspective, major identity policy issues in relation to Cultural Christianity that can be mobilized for illiberal purposes concern migration control with regard to people coming from predominantly Muslim countries. A related but distinct issue concerns the gender equality frame. The third aspect, which the remainder of this section briefly discusses, and which Joppke and Astor and Mayrl emphasize, is the national heritage frame in the context of court decisions.

### 3.2.1. "Othering": The Illiberal Christian-Occident Frame

Considering the ambivalent effects of secularization and rapid sociocultural change since the 1980s, religion has returned to radical-right mobilization, even in secularized societies (Minkenberg 2018, p. 34). Cultural Christianity's vulnerability to illiberal aims comes into play if one situates the idea of a Christian nation in the context of the New Right's concept of "ethnopluralism", a countermodel to concepts of multiculturalism. Ethnopluralism "emphasizes the incompatibility of cultures and ethnicities and advocates

the right of the Europeans to be different, to preserve the cultural (Christian) identity of the nation, and to resist cultural mixing" (Minkenberg 2018, p. 9). How Christianity becomes mobilized in this context is exemplified by the PEGIDA movement, if only by its featuring "Occident" in its name and hauling a wooden cross through the streets. It is likewise with the slogan of the right-wing populist Freedom Party of Austria, "Abendland in Christenhand", or "the Occident in Christian hands" (Joppke 2018, p. 5).

This is the practice of "othering". From a postcolonial perspective, it is an old strategy of the West, and of Europe in particular, used to create an exclusive and hegemonic counteridentity to what is construed as "the Orient" (Said 1985). The recourse to religion or a construed Christian tradition helps "to create an epistemic, political, anthropological and thus quasi-ontological border between a—more or less explicit—"we-group" and "the others" and to objectify it" (Mecheril and Thomas-Olalde 2011, p. 46). Nowadays, religious othering takes place in the context of globalization and migration, a phenomenon that Storm (2011), Marzouki et al. (2016), Lazaridis and Campani (2016), and Hennig and Weiberg-Salzmann (2021) also observe. Case studies in these volumes reveal how right-wing populists hijack or rent Christian religion to mobilize a Christian national identity in opposition to the *idea* of a pluralistic society, an idea deeply contested in recent years, especially in light of immigration from predominantly Muslim countries. This happens despite the fact that there "are no major 'religious' or Christian-affiliated political parties in Europe, and political decision-making is increasingly independent from religion". It is because "the issue of 'identity' has in recent decades risen to the top of political debate" (Roy 2016, p. 200).

Analyses of right-wing parties in Italy, Austria, and Hungary demonstrate that the illiberal appropriation of Cultural Christianity occurred over time. Political actors or programs underwent a steady process of ideological-strategical transformation. Like Orbán in Hungary, for instance, or the Lega Nord in Italy, they established themselves as secular actors, only to then discover the potential of Christian narratives as a means of excluding "the other" (Bolzonar 2021). Unlike the concept of Christianism (Brubaker 2017; Ryan 2021), our approach to Cultural Christianity allows us to argue that references to an illiberal Christian Occidental frame, here considered an expression of an illiberal appropriation of Cultural Christianity, are not limited to political actors. Conservative or nationalist-minded church representatives may also make use of an anti-Muslim reference to Christian traditions and values, as in Poland where Cultural Christianity has been leveraged as a claim against an open asylum policy (Hennig and Resende 2021).

### 3.2.2. The Illiberal Gender Frame

The second frame can be seen as a variation of this "othering". It refers, with Brubaker (2017, p. 1210), to the defense of gender equality among right-wing actors, but is undertaken on the basis of exclusionary motives, especially in liberal democracies where liberal principles of equality and individual rights are widely accepted. He sees a particular pattern in Western and Northern Europe where gender equality is framed as a characteristic national value. Here, "Christianity is redefined as the matrix of liberalism, secularity, gender equality, and gay rights", which right-wing parties apply against Islam as the "illiberal other". Gender equality turns into a European civilizational value with roots in the Christian tradition, while gender inequality and oppression are represented as inherent in Islam. "Muslim women are represented as victims of enforced covering, forced marriages, spousal violence, polygamy, genital cutting, and honor killings, while Western women are represented as threatened by conversion as well as by sexual assault from Muslim men" (Brubaker 2017, p. 1203). Accordingly, the culturalization of religion is convenient from a nationalist-populist point of view also because "it allows Muslim religious practices, redefined as cultural, to be restricted in a way that would not otherwise be possible, given the liberal state's commitment to religious freedom" (Brubaker 2017, p. 1210).

In Central and Eastern Europe, gender equality politics are viewed more often than not as an import, or an imposed doctrine from Brussels, and less as a canonical ingredient of European values. It is more likely that political or religious right-wing populist actors will refer to Europe's Christian foundations and values as a way of counteracting the institutionalization of LGBT+ and women's reproductive rights (Graff and Korolczuk 2018). In Russia, the Orthodox Church has become a publicly influential player, despite rather low church attendance, where a growing emphasis on traditional moral values, and the introduction of new holidays dedicated to celebrating the traditional heterosexual family (Köllner 2021), may also display a reference to a sort of Culturalized Christianity.

### 3.2.3. The Illiberal Heritage Frame

The final frame on which political, religious, and societal actors rely, and which displays the susceptibility of Cultural Christianity to illiberal use from the perspective of actors and their agendas, connects to the observation that "references to religious heritage have become increasingly common in political discourse, judicial rulings, and public discussions in Western societies". This sort of reinforced identification of "post-Christian" societies with church buildings or Christian symbols, such as the crucifix, seems, but is not necessarily, triggered by the affirmation of migrants' non-Christian identity and its presence in the public sphere. This sort of reconsideration of Christianity as part of a cultural-national identity can also display a fearful or racist reaction to the presence of Islam in Europe (Burchardt 2020, p. 155f). The situation in Russia and Central Asia is different in this regard since the systematic restitution of church buildings to religious communities has only begun very recently. Research on these dynamics is still ongoing (Köllner 2018).

One example is the legal struggles over the legitimacy of religious symbols, such as the crucifix or the veil, in the public sphere. Struggles like these may encourage courts to reconcile the privileged position of majoritarian religions with liberal principles of secularity and neutrality. A decision that legitimates the presence of a crucifix in public offices or classrooms on the grounds that it is a symbol of cultural heritage, rather than a religious symbol, could set a precedent that would allow women to wear a hijab in public offices, for instance, since that too, it could be argued, represents a cultural symbol. Another culturalized interpretation, however, considers the hijab a political symbol and provides, on this basis, a frame for rejecting the right to wear it in schools, for instance (Reichelt 2020). Such references to a religion-as-heritage frame (Burchardt 2020, p. 157) may be mobilized by religious and political actors alike. Thus, "those making pragmatic use of religious culture can draw upon culturally religious identities as a resource to build support for their struggles". On the other hand, "activists may appeal more directly to these identities for political purposes" (Astor and Mayrl 2020, p. 221).

### 4. Selective Memory Politics: The Idea of Christian Europe as an Exclusivist Identity Marker

Section 3 suggested two major approaches which can help us to better understand the vulnerability of Cultural Christianity to illiberal appropriation. The final section explores the idea of Christian Europe as a selective, potentially exclusivist, and transnationalist identity marker[10] that is supported by the superficiality and dogmatic flexibility of culturalized religion. Taking the idea of Christian Europe as an example, our approach then draws on a historical-theoretical perspective to examine the changing role that the idea of Europe, or the European identity, plays within the political doctrine of contemporary right-wing parties and actors. In this respect, we ought not forget that the (secular) idea of the "nation", which traditionally dominates right-wing constructions of collective linguistically, culturally, and ethnically homogeneous identities, historically emerged in clear opposition to the traditional Christian conceptions of politics. More precisely, it was the secular doctrine of sovereignty (Bodin, Hobbes), as well as the intellectual emancipation of the political from religious guidelines (Machiavelli, Botero), that gave rise to the concept of

the modern and secular nation-state. The creation of the intellectual environment necessary to support this is due, in no small part, to the loss of power the Catholic church suffered through the Reformation, and the subsequent increase of religious plurality. Thereafter, the focus of political power steered away from the universalist claims of the Christian religion (or, more precisely, Roman Catholicism) during the medieval period, towards territorial rule over a nation (Hidalgo 2019a).

### 4.1. (Territorial) Nation vs. (Universalist) Religion? Structural Differences and Analogies

In the context of modern Europe, the idea of the nation advanced. It became both the guarantor of, and a quasi-synonym for, a social and political unity, without which a state or a people, in the narrower sense, would have been inconceivable. Hence, with the slogan, "What is a country, if it is not a nation?", Clifford Geertz (1997) once summarized this particular understanding of the European state, which treats the nation as a significant and exclusive political association. In addition, the secular background to this intellectual history was elaborated by Benedict Anderson. As his remarkable study *Imagined Communities* (Anderson 1991) points out, the modern concept of the "nation" is not only "limited" or defined by its territorial borders,[11] but it also has to be understood as reflecting a deep contrast with the earlier religious communities that were connected by "sacred languages" (Anderson 1991, p. 13ff.). For Anderson, it would be an exaggeration to say that the "appearance of nationalism was [directly] 'produced' by the erosion of religious certainties" (Ibid., 10), or that the attainment of nationalism meant an overcoming or replacement of traditional religions. Indeed, he highlights the fact that the concept of the nation primarily came "to maturity at a stage of human history when even the most devout adherents of any universal religion were inescapably confronted with the living pluralism of such religions, and the allomorphism between each faith's ontological claims and territorial stretch". Likewise, the nation "is imagined as *sovereign* because the concept was born in an age in which Enlightenment and Revolution were destroying the legitimacy of the divinely-ordained, hierarchical dynastic realm" (Ibid., 7). In this particular historical situation, over the course of the 17th and 18th centuries, the modern national state demonstrated its sovereignty by becoming independent from religion. Thus, it was apparently no coincidence that "in Western Europe the 18th century marks not only the dawn of the age of nationalism but the dusk of religious modes of thought" (Ibid., 11).

However, the contrast between religious communities and secular nations describes only one side of the coin. The other side is the structural similarity that obviously exists between them. According to Eugen Weber (1991, p. 23), "history was the theology of the 19th century because it provided societies cast loose from custom and habit with new anchorage in a rediscovered past". The resulting historical myths could thus serve as a basis for framing national memories that benefit the particular goals of certain political associations. Anderson himself suggests a comparable nexus between religious communities and nations by identifying both as "cultural systems". In this respect, nation-states are both historical products of a culture shaped by "rationalist secularism" *and* the "political expressions" of imaginations, "which [ . . . ] always loom out of an immemorial past and, still more important, glide into a limitless future. It is the magic of nationalism to turn chance into destiny" (Ibid., 11f.). With this perspective, Anderson is very close to the work of Ernest Renan, who analyzed the "dynamics of national consciousness" as a double process of "memory" and "forgetting" (Cheng 1995, p. 215).[12] As early as 1882, in his famous speech at the Sorbonne,[13] Renan described the concept of the "nation" as a kind of spiritual substitute for religion. Accordingly, the past, present, and future shared by a people is supposed to be understood as the constitutive *principe spirituel* of the nation. For Renan, it is not a common language, a general interest, or a geographical location that is decisive when a nation is formed, but the collective memories of the past and the desire to feel like a political unit, not only in the present but on into the future (Renan 1996). Correspondingly, Clifford Geertz (1997) identifies the original source of the nation with the primordial ties of a group, perceived as "supra-temporal", while Klaus P. Hansen (2009,

p. 149ff.) recognizes the nation as a synonym for community in general, and no less than a "collective of destiny".

### 4.2. Culturalized Religion and (Trans-)Nationalist Propaganda

Against the background of the ideological history of the "nation", it should become evident why culturalized religion in general, and Cultural Christianity in particular, are such game-changers with respect to the formation of a specific "religious" right-wing nationalism. As we have conceptualized in Section 2, Cultural Christianity can be seen both as a dilution of traditional religious dogmas, including a detachment of a (pseudo-)religious identity from (universalist) theological claims, and as a resurgence, or even reinforcement, of exclusivist interpretations of Christian heritage. With regard to the relationship between nationalism and religion that is relevant here, this sort of culturalized Christianity is particularly capable of adapting to the territorial logic of national and secular domination and to a spatially broader form of collective identity across and beyond traditional national borders. This paradoxical amalgamation between right-wing nationalist and culturalized religious attitudes is apparently favored by the fact that Cultural Christianity refers to a form of religion that, on the one hand, shows only a superficial presence in secular society, but that, on the other hand, increases its potential to demarcate a religious-national in-group from a relevant out-group precisely because of this. Accordingly, following Anderson and Renan, the merely superficial presence of Christian dogmas in secular society helps us to *forget* that there was once a very fruitful historical relationship between Christians and Muslims in terms of philosophy, science, and culture,[14] while collective *memory* can easily reactivate and reanimate the (rather salient) historical-political competitions between them in the present.[15]

Proceeding from this, Cultural Christianity can simultaneously act as an identity narrative for national, transnational, and post-national memory politics in historical and contemporary Europe (cf. Medina 2018). Illiberal, right-wing actors in particular can activate this potential in order to construct collective identity narratives in sharp contrast to Muslim traditions and the idea of a multicultural society. Through culturalized and secularized forms of (Christian) religion, an intersection of religion and memory politics can, in principle, be restored, thus shaping a 'Western' or Occidental collective identity against all so-called Oriental influences. However, although this resurgence of (secular) religion as a resource for (European) memory politics indicates, at least to some extent, the new rise of a "postnational", culturo-Christian identity in Europe, the popular invocation of the moral and cultural heritage of Christianity must nevertheless be seen as the result of a nationalist or even chauvinistic agenda. Since Islamophobia has become the most dominant ingredient within current right-wing ideologies,[16] these sorts of nationalist policies simply call for a (postsecular) reassessment of the role of Christianity in European history and memory. In other words, in order to strengthen a collective social identity against the transnational "out-group" of Muslim immigrants, the relevant "in-group" can hardly be shaped by a political ideology that is merely nationally oriented.

### 4.3. The Idea of Christian Europe within (Trans)Nationalist Memory Politics

Paradoxically, memory politics for nationalist purposes today benefit from the (post-national and postcolonial) narrative opposition between Orient and Occident, and Islam and Christianity, and tends to change the image of Europe from a supranational union to a coalition of Christian nations. At the same time, the narrative of Cultural Christianity becomes apparently capable of arousing feelings of hatred or fear against Muslim refugees and immigrants, thus reinforcing their political and social exclusion.

The relations and references just outlined are all the more noticeable as they obviously break with the pejorative image of the European Union that had previously circulated among radical right-wing actors and voters. In all likelihood, such a significant course correction on the radical right became conceivable after the enemy image of Islam became established and motivated nationalists all over Europe to seek illiberal recourse in (postna-

tional) Christian culture. Conversely, this means that the illiberal instrumentalization, or "hijacking", of Cultural Christianity that we have been discussing could only happen under the condition that a highly selective memory politics substantially redefined the historical role of the Christian religion, which was actually directed *against* the modern and secular idea of the nation. Accordingly, we are able to observe two communicating vessels: illiberal Islamophobia motivates a selective recourse to Christianity, and the selective reading of Christianity is, in turn, a prerequisite for the instrumentalization of Christianity in order for the idea of illiberal Islamophobia to succeed at all.

Such an illiberal memory politics (and memory politics is closely linked to the construction of a national identity, see Wolfrum 1999) has adapted the relics of Cultural Christianity in a very one-sided manner in order to reanimate the historical opposition between two premodern political-religious empires: the Christian Occident and the Muslim Orient. This narrative strategy transvalues the traditional Christian idea of Europe in order to suit the requirements of a nationalist right-wing agenda. As we have seen, it also provides the appropriate pattern for an "anti-Muslim identity" beyond national borders, since a single nation-state or nationalism would hardly be a relevant category to oppose a (supranational) religious community such as Islam.

In recent years, this central aspect has inspired right-wing populist (or extremist) movements to enhance their traditional veneration of secular ideals, such as the "nation" or the "racial corpus", with an explicitly religious invocation of the "cultural greatness" of the Christian Occident (Minkenberg 2018). As a result, not only has the enemy image of the European Union receded into the background as the hitherto dominant identity resource for right-wing actors in Western and Eastern Europe, but radical right-wing groups such as the Identitarian Movement have even begun to form themselves as a pan-European project *beyond* the EU (Virchow 2015; Zúquete 2018).

The selectiveness of memory, which is inherent to the concept of the "Christian West" as an alleged supranational identity of (right-wing) Europeans, is demonstrated, moreover, by the real cultural struggles that once raged between Enlightenment thinkers and church representatives, and modernists and conservatives, in modern Europe. In this respect, contemporary memory politics effectively conceals the originally massive resistance of Christian churches to modern achievements, such as democracy, human rights, and religious freedom. It seems all the more questionable, then, to suggest that the historical processes of secularization and the separation of religion and politics are exclusive achievements of the Christian or Western world.[17] However, the right-wing rhetoric of wanting to defend the Occident against the current "Muslim invasion"—which has been articulated and embodied most unequivocally and rigorously by anti-immigration and Islamophobic right-wing actors such as Viktor Orbán, Matteo Salvini, Marine Le Pen, and Heinz-Christian Strache—is gaining in persuasive power among large sections of European societies, since it obviously fulfils a widespread need for collective identity today (Hidalgo 2019b). At the same time, the political agenda of Cultural Christianity in this form could even lead to a Europe-wide alliance of right-wing parties,[18] which would be inconceivable from the outset without the common enemy image of "Islam".

*4.4. A Clash of Christian and Muslim Civilizations?*

Facing the deep identity crisis into which contemporary Europe has plunged as a result of the social and ecological downsides of globalization, the ageing of European societies, and worldwide migration movements, right-wing actors' observable recourse to Cultural Christianity is thus more than a sideshow designed to generate more votes, even if this certainly plays a role. In tandem with the negative image of Islam, the category of religion has advanced as such to become a central feature in the process of collective identity formation, as far from (measurable) criteria, such as church affiliation and/or concrete religious practice, as it may be. Furthermore, the religious-secular identity offered by Cultural Christianity could all the more easily provoke persistent group polarization, since radical Islamists are willingly taking up the so-called division between the Western and

the Islamic world. In their very own way, they are claiming that Muslims are generally not welcome and extremely discriminated against in European societies, which prompts them to call for revenge and violence instead of integration or even assimilation. Consequently, the pronounced "clash" of Christian and Muslim civilizations is in danger of becoming a kind of self-fulfilling prophecy because whichever is treated as an enemy will effectively become one at some point, without the need for any further cause.

In addition, the desire to create a positive image of one's own religious culture, and to defend it against other cultures (or against multiculturalism), ultimately arises from one of the basic motivations in human life, namely, the experience of risk and insecurity. Thus, confidence in the superiority of one's group identity, as well as the subjective conviction that one is part of a significant, permanent, and, if you will, "immortal" entity such as a culture, a nation, or a religion helps in coping with one's inevitable mortality. In turn, the collective feeling that one's own culture is threatened by subjugation or infiltration makes one's own mortality all the more apparent.

This thesis, which is well-known as the Terror Management Theory (Greenberg et al. 2008), is able to explain why the *symbolic* threat potential of "Islam"—i.e., the associated fear of the "decline of the West" (Oswald Spengler)—is usually considered far more serious than the *actual* risk of becoming the victim of an Islamist terrorist attack. Moreover, the same threat also explains why many (migration-critical) people continue to confuse integration with assimilation (cf. Koopmans 2017), simply because the latter would obviously be a kind of "symbolic" victory over the "others", as well as a performative confirmation of the superiority of one's own cultural values.[19]

In Europe's current social and political environment, which is characterized by numerous fears and rather gloomy prospects for the future, such religiously impregnated right-wing illiberalism has a relatively easy job of satisfying the growing demand for social identity and symbolic superiority. Hence, characteristics, such as the antipluralistic insistence on social homogeneity qua exclusion and discrimination against minorities (Müller 2017), or socio-cultural authoritarianism (Mudde 2007), are ciphers to which a right-wing instrumentalization of (Christian) religion, such as Cultural Christianity, comprehensively contributes. The issue of religion, or Cultural Christianity, thus ought to be seen as a key question in the political confrontation with right-wing illiberalism and radicalism.

## 5. Conclusions: The Illiberal Vulnerability of Cultural Christianity

The aim of this paper has been to explain why Cultural Christianity, as a rather neutral expression of secularized religion in a postsecular society, is also appropriated for illiberal purposes. To that end, we have argued that the persistence of a non-faith-based reference to Christianity has to be seen in the context of the ambivalent nature of secularization, which in turn is an expression of the ambivalences of modernity in Europe and beyond. To explore how Cultural Christianity is susceptible to illiberal and postnationalist propaganda, we have proposed two standard perspectives: first, the demand side, which entails looking at data in order to understand to what extent we can observe linkages between a nonreligious self-identification with Christianity and illiberal ideologies, such as nationalism or Islamophobia; and, second, the supply-side perspective within a market where identity "products" are on the rise, which are essential ingredients for power-seeking political actors. Here, we have identified three major frames through which Cultural Christianity is mobilized for illiberal purposes: The *illiberal Christian-Occident frame, the illiberal gender frame,* and *the illiberal heritage frame.* Whereas these three frames represent a classical empirical social scientific approach, the third perspective involves a deeper historical-theoretical analysis of the "illiberal Occident frame" along the lines of "Europe" as a historically somewhat paradoxical example of selective identity constructions. In this respect, the last section has shown that the characteristics of Cultural Christianity—its religious superficiality and the detachment of a pseudo-religious identity from universalist theological claims—facilitate the resurgence of exclusivist interpretations of Christian heritage by linking the actual con-

tradictory logic of national and secular domination with the transnational collective identity of Christian Europe. At the same time, this helps to explain why Cultural Christianity remains susceptible to a form of illiberal instrumentalization.

In conclusion, we have argued that, given the context of religiously and culturally pluralizing European societies, the demand for, and mobilization of, an exclusivist Christian identity from the right needs to be carefully analyzed. However, if one recalls the vibrant controversy within the European Union surrounding the inclusion of a reference to Christianity in its never-realized draft constitution in 2003 (Joppke 2018, p. 4), it is reasonable to assume that the idea of a Judeo-Christian Europe may be accepted in the political center as well. At the same time, more data about "believing without belonging", and its potential linkage to illiberal visions, would be needed to better estimate the "real" potential of illiberal identity mobilization by way of a Cultural Christian frame.

In terms of future research, we believe that at least two relevant and interrelated questions require further in-depth investigation. It seems true that illiberal narratives building upon Cultural Christianity are most frequent in societies with low religiosity and (with the exception of Eastern Germany and Hungary) without a Communist past. However, in other post-Communist contexts, like Poland, which is still very religious, a more nuanced perspective is needed to understand, for instance, to what extent the presence of Polish Catholic nationalism is a cultural or a religious one, or to what extent the new public presence of, and reference to, the Orthodox church in Russia may be more or less cultural or religious (Köllner 2021). Relatedly, another open question concerns whether any references to the above-mentioned frames by religious representatives themselves could in any way be deemed an aspect of Cultural Christianity.

**Author Contributions:** Both the authors made equal contributions to the content of the article; funding acquisition, A.H. All authors have read and agreed to the published version of the manuscript.

**Funding:** The authors would like to thank the Institute for European Studies (IFES) of the European University Viadrina for funding the language editing of this publication. Additionally, we acknowledge support from the Open Access Publication Fund of the University of Münster.

**Conflicts of Interest:** The authors declare no conflict of interest.

## Notes

[1] For the concept of "civilized religion" [*zivilisierte Religion*] within the liberal state, see Schieder (2008).

[2] On the notion of "Christianism" as an option to characterize the use of Christianity by right-wing populists in order to advance their nationalist ideological agenda, see also Ryan (2018, 2019, 2021).

[3] Europeanization is here understood as a process towards converging political values, politics, and experiences already prior to EU-related policy change.

[4] Similar to Hervieu-Léger, Meulemann understands "Kulturchristentum" not as a new "Angebot" (offering or sales pitch) to nonreligious people but as a result of secularization processes through which people are able to share in a positive view of the church as an institution, to celebrate religious holidays, and even perhaps organize church marriages or funerals though they are no longer believers in the church.

[5] The problem begins, according to the Bertelsmann Foundation, with a lack at present "of reliable, up-to-date data on religious affiliation in many European countries", as shown by the Swiss Metadatabase of Religious Affiliation in Europe (SMRE). In addition, many of the statistics relating to religion would contradict one another, "possibly because the question of religious affiliation is sometimes raised objectively, sometimes subjectively" (El-Menouar 2017).

[6] The percentage of affiliated and un-affiliated Christians in select West European countries is as follows: church-attending Christians: Austria (28%), France (19%), Germany (22%), Italy (40%); nonpracticing Christians: Austria (52%), France (46%), Germany (49%), Italy (40%) (PEW 2018a, May 29).

[7] In select countries surveyed, such as Greece (89%), Russia (69%), Romania (66%), Poland (55%), Hungary (46%), and even the Czech Republic (55%) (PEW 2017, October 5, p. 150).

[8] See Brubaker's contribution to the Immanent Frame Blog: https://tif.ssrc.org/2016/10/11/a-new-christianist-secularism-in-europe/ (accessed on 14 June 2021).

[9] In mostly Orthodox countries, a large majority of people say their culture is superior to others: in Romania 66%, Bulgaria 69%, Russia 69%, and in Greece 89% (PEW 2017, October 5, p. 150).

10    For the concept of transnationalism as a politics of belonging under the conditions of globalization, see Westwood and Phizacklea (2000).

11    "No nation imagines itself coterminous with mankind. The most messianic nationalists do not dream of the day when all the members of the human race will join their nation in the way that was possible, in certain epochs, for, say, Christians to dream of a wholly Christian planet." (Anderson 1991, p. 7)

12    For this, Anderson (1991, p. 6) quotes probably the most famous sentence from Renan's *Qu'est-ce qu'une nation?*: "l'essence d'une nation est que tous les individus aient beaucoup de choses en commun, et aussi que tous aient oublié bien des choses" "the essential element of a nation is that all its individuals must have many things in common but it must also have forgotten many things" (Renan 1996, p. 42). As an example of the latter, Renan mentions the St. Bartholomew's Day Massacre in 1572.

13    See the previous footnote.

14    See, e.g., Armstrong (1993) and Smith (1999).

15    See Lewis (1993).

16    See Section 3.

17    For this argument, see Huntington (1996).

18    https://www.theguardian.com/world/2019/may/02/matteo-salvini-vote-for-nationalist-parties-stop-islamic-caliphate (accessed on 20 April 2021).

19    For this, we invoke Charles Taylor (1992, p. 38) and his perception of the politics of difference as the recognition of "the unique identity of [an] individual or [a] group, their distinctness from everyone else", which should not be "assimilated to a dominant or majority identity".

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
