# Peer review of "Illiberal Cultural Christianity? European Identity Constructions and Anti-Muslim Politics"

_religions, doi:10.3390/rel12090774_

Round 1

Reviewer 1 Report

“Illiberal Cultural Christianity? European Identity Constructions and Anti-Muslim Politics” is a very timely, interesting, and thought-provoking paper that provides an innovative interpretation of Cultural Christianity. The manuscript is well-written, and the general organization of the paper is quite sound. I have just a few suggestions for clarifying and deepening certain aspects of the argument.

The question of directionality is an important methodological challenge that complicates analyses of the relation between Cultural Christianity and Islamophobia / xenophobia. Are people who are Culturally Catholic more likely to feel antipathy toward Muslims, or does feeling antipathy toward Muslims lead people to identify as Culturally Catholic as a means of distinguishing themselves from the Other? Or perhaps there is a third factor that accounts for both? The authors seem to favor the interpretation that anxiety about Islam leads people to re-examine their religious identities and cite cross-sectional data, but such data cannot really answer this question. The authors should cite Ingrid Storm’s “Christian nations? Ethnic Christianity and anti-immigration attitudes in four Western European countries” on this question.

Core to the authors’ argument is the idea that secularization is ‘ambivalent’. I would urge them to reconsider this term. Ambivalence connotes mixed emotions. It is hard to see how something abstract like secularization can be ambivalent. Perhaps ‘ambiguous’ is what they have in mind? Or terms like ‘multi-faceted’ or ‘multi-dimensional’ might work. The key is that public religious identities are growing in salience despite a general decline in religiosity (belief and practice). The term ‘ambivalence’ does not really capture this paradox, or if it does, it must be explained better.

The authors reference insecurity as a reason people valorize their own religious culture over others. This is important for explaining the ‘demand’ side of the argument for why Cultural Christianity is proliferating. An alternative explanation comes from Voas, who conceives it an ordinary (and less politically-motivated) phase of the secularization process. The authors might think about explaining how their argument coheres or conflicts with this account.

Althusser’s notion of ‘interpellation’ might be useful to incorporate as a means of clarifying the socio-psychological mechanisms at play. When conservative or far right parties invoke the Christian heritage of the nation, they are in a sense interpellating potential supporters at Christian subjects, even the non-observant. Many of these supporters are pre-disposed to recognize themselves as Culturally Christian for the reasons discussed by the authors.

The authors comment on a couple occasions that exclusionary nationalist rhetoric is insufficient for justifying the exclusion of a transnational outgroup like Muslims. I wonder if this is really the case. Consider, for example, Braunstein’s article, “Muslims as Outsiders, Enemies, and Others,” which discusses different framings of Muslims as anti-American, un-American, non-American, etc. While ‘American values’ are often framed as universal, it is not clear that nationalist rhetoric is somehow inadequate for articulating exclusionary positions toward transnational religious groups. In fact, there is a long history of this around the world.

Author Response

Thank you very much for your helpful comments to our paper.

  • We think, that anxiety about Islam is multidirectional, however, literature we used suggest the construction of a Cultural Christian identity rather as reaction to Muslim immigration and identity building. The question is rather, whether fears of Muslims is a product of the radical right.
  • We included a sentence with a reference to Ingrid Storm
  • We use the concept of “ambivalence” not from an empirical but theoretical perspective. While also “ambiguous” would fit, following Zygmunt Baumann, the reference point is modernity and the idea of ambivalences of modernity. We prefer to stick to that term as it links to a new and promising research perspective.
  • We are greatful for the advice concerning Voas and the alternative argument for why Cultural Christianity is proliferating but we fear that including a further perspective would rather confuse than enlighten the reader. Apart from that, we think that both arguments are complementary
  • The same is to say about the very interesting suggestion concerning Althusser and the notion of interpellation. It would be very interesting to deepen and specify the question of why Cultural Christianity can be invoked. That could be an interesting following-up paper. But we see no room for including socio-psychological mechanisms beyond an only cosmetic manner.
  • Finally, the first reviewer is right, of course, that we pursue a European-centric perspective that does not include the US-American discourse. It would be interesting to compare these different contexts, with the US as, compared to Europe, a still very religious society. However, for this article the focus on “secular” Europe is on purpose.

Reviewer 2 Report

The paper proposes an interesting angle in the study of religion and discrimination discourses (broadly understood). It engages, more specifically, with the use of religion by nationalists for illiberal purposes (p.1, lines 32-33). The key focus is the theoretical exploration from a political science perspective of the appropriation of christianity in Europe towards illiberal use, in particular in construction of the contrast with the Muslim religion as 'Other'.

The article is overall soundly structured and presented in a coherent way.  There is, additionally, reasonable use of relevant literature. There are, nevertheless, a few points where more clarity would be helpful for readers (who given the topic of the article are likely to come from different social science disciplines):

  • p.4 - lines 178-179: the authors claim that self-identification with Christianity without being observant is a marker of the ambivalence of modernity and Europeanization. The connection of religious identity with Europeanization/modernity is not very clear here and perhaps it would be worthy of being spelt out.
  • for section 1.2 : the conceptual use of secularisation was not clear to me. Are we to understand its use as a political concept denoting the separation of state from church or are we rather to interpret it as the distancing of people from religion? This would matter as it would allow us to 'see through' the nature of the transformation of the uses of Christianity in the political discourse (see also other comments below)
  • linked to the previous comment, the conceptualisation of cultural christianity as an outcome of secularisation is probably the weakest argument of the paper. If secularisation is not conceptually clear (page 5- lines 226-227 for example), how can we claim that it is at the source of the illiberal uses of christianity by nationalists? Put differently, one could for example claim that the 'window of opportunity' to use christianity in the way that is outlined in the paper, is the result of the fact that now Europeans believe in more fragmented and eclectic ways, picking and choosing (e.g. younger generation of Catholics in Poland) but they still 'believe' (so the opposite of secularisation in its traditional sense of decline of religion). 
  • I was also not clear of how the concept of 'exculturalization' discussed on page 5 from line 246 onwards helps the general argument.
  • Thinking of the link between cultural religion and far-right identity politics (page 10, line 442 onwards), it might be oversimplified to claim that there is continuity when it comes to the cultural religion and its close ties with ethnicity, nationality and social class. In Germany, since PEGIDA is mentioned a number of times in the paper, religion is mostly constructed as a marker in the context of immigration and much less a generational process, unless the authors here mean something different.
  • The argument about demand and supply for identity 'products' (page 16, lines 765 et seq), leaves completely out the role of populism as a 'broker' for the creation of demand. In fact, while the 'shadows' of populism exist is several parts of the paper, the instrumentalization of religion in cultural terms is largely the product of the quest for political power at its core. 
  • a smaller issue: on page 16, lines 747-750: this statement requires some nuance and if not, references to back it.
  • Similarly, lines 756-757 are also not clear in their meaning (i.e. what is there to lose and what is there to win?)
  • The statement that 'the idea of Judeo-Christian Europe may be accepted in the political centre as well' (line 787, page 16) makes me wonder whether it is rather a strategy to attract votes rather than an ideological conviction/change in the way one believes, in which case the issue of populism enters again in the picture of appropriation of christianity by political parties.
  • Finally, one last point that occurred in the last paragraph of the paper (lines 791 onwards) when you refer to the role of religious actors in the use of cultural religion: ultimately can we really not say that cultural christianity and cultural religion are more political and less religious, even when they come from religious actors themselves? 

Author Response

Thank you very much for your helpful comments.

  • With regard to lines 178-179, our sense it that readers perceive “Europeanization” as a concept necessarily related to EU-politics. We included a footnote to clarify that, arguing that Europeanization has been happening long before EU-Europeanization.
  • I 2: We see secularization as a political concept with a lot of different meanings. Both aspects the reviewer has mentioned – the separation of state from church and the general distancing of people from religion – are relevant for us, but both aspects are nevertheless contested, if you think for example of the German case where the institutional separation between church and state is not really given or the cases of USA, Brazil and Spain used by José Casanova to prove his thesis that secularization does not necessarily mean a decline of piety and devotion or also a privatization of religious beliefs. Therefore, we are very aware of using an alternative concept of secularisation in order to emphasize the common aspects of its different meanings yet.
  • We do not claim that secularization is a source for the illiberal use but that it is a relevant condition for Cultural Christianity coming into existence. The fact that CC is vulnerable and can be appropriated for illiberal purposes has to do with the idea of modernity (and, thus, also secularization processes) being ambivalent. We think that we made that clear.
  • We agree, that one may conclude that Europeans still believe, although they do it in more fragmented and eclectic ways, and that this is against an idea of secularization in its traditional sense as decline of religion. However, we don't see any contradiction to the concept of cultural christianity outlined in the paper, since we do not associate secularization with decline of religion but with Taylor as the coexistence and plurality of religious and non-religious lifestyles and values. Furthermore, the point the reviewer made is part of the transformation of the religious field as such and of “liberal” Cultural Christians as well. But our focus is merely on the illiberal side of this process.
  • The concept of exculturalization links to the perspective of “ambivalence” within the relationship between culture and religion under the condition of globalization (modernization) processes
  • Unfortunately, we have not understood properly the point concerning line 442 and PEGIDA.
  • The role of populism as a 'broker' for the creation of demand was not excluded from our argument, as we referred to right-wing parties. But it was a good idea by the reviewer to emphasize that power-politics is a driving factor; so we added it to the conclusion. 
  • Lines 747-750: As a reference, we have added a footnote quoting Charles Taylor’s critique on assimilation and the distinctness of an individual or a group as main characteristic of identity.
  • Lines 756-757: We have deleted this last sentence of section 3 because it really had no meaningful content. Thanks for this advice.
  • It is both and interdependent, the idea of interconnected supply and demand-side factors try to grasp this dynamic theoretically. But it is difficult to trace such dynamics empirically.
  • The appeal to Judeo-Christian Europe is both a strategy to attract votes and an ideological conviction, and both aspects are interdependent. In this concern, the idea of interconnected supply and demand-side factors tries to grasp this dynamic theoretically. But it is difficult to trace such dynamics empirically.
  • Finally, it is a very difficult question if we can still say that cultural christianity and cultural religion are more political and less religious, even when they come from religious actors themselves. That is also the reason why we have reduced complexity and did not include the ambiguous role of religious actors into our analysis but left it for further research. On the other hand, one could also argue that religious actors who use cultural religion for political purposes are predominantly political actors anyway. This is once again an advantage of our concept of secularization which allows to grasp the deep intertwinement of politics and religion under secular circumstances. 

Reviewer 3 Report

As a whole, the manuscript has a clear and logical structure, and it is not difficult to follow what the authors want to say. In my opinion, the authors have sufficiently used the relevant and recent bibliography. The title and the abstract correspond to the content of the paper. In the beginning, there is a clear statement about the perspective, and the paper's goal, including the overview of research, conducted hitherto (the state of the problem), including the relevant literature. After that, the authors explain their concepts of Cultural Christianity and the Illiberal within the frame of the ambivalent secularization (the understanding of it is defined too, including the relevant references with different approaches to the concepts and clear statements about their position). The second section is about why and how Cultural Christianity (neutral concept) and the Christian religion can be corrupted and misused by illiberal politics. That happens, from the sociological perspective, within three frames (Christian-Occident, gender, heritage). The specific contribution of the paper is the inclusion of the historical-theoretical view (within the heritage-frame) of Cultural Christianity with the claim that Cultural Christianity is an outcome of secularization.

All claims are supported with references, and the authors' view is expressed clear and distinctive, especially concerning the ambivalence of Cultural Christianity. In Conclusion, the Authors stress two open questions, which are the consequence of lacking research of these topic perspectives. That notion shows their awareness that there are some elements that can contribute to a complete picture of the topic. The manuscript was interesting to read, and I look forward to the continuation to answer the questions raised in the Conclusion.

Author Response

  • Thank you very much for commenting our paper so positively. We are also looking forward to continue to answer the questions raised in the conclusion. The question concerning Cultural Christianity and its instrumentalization by illiberal actors will keep us busy in the future.